# A Novel Localization System in SAR-Demining Applications Using Invariant Radar Channel Fingerprints

**DOI:** 10.3390/s22228688

**Published:** 2022-11-10

**Authors:** Nicholas Karsch, Hendrik Schulte, Thomas Musch, Christoph Baer

**Affiliations:** Faculty of Electrical Engineering and Information Technology, Institute of Electronic Circuits, Ruhr University Bochum, Universitaetsstr. 150, 44801 Bochum, Germany

**Keywords:** RF fingerprints, channel identification, frequency coding, FMCW radar, localization

## Abstract

In this paper, we present a novel two dimensional (2D) frequency-modulated continuous-wave (FMCW) localization method for handheld systems based on the extraction of distinguishable subchannel fingerprints. Compared with other concepts, only one subdivided radar source channel is needed in order to instantly map a one-dimensional measurement to higher-dimensional space coordinates. The additional information of the detected target is implemented with low-cost hardware component features, which exhibit distinguishable space-dependent fingerprint codes. Using the given a priori information of the hardware thus leads to a universally applicable extension for low-cost synthetic aperture radar (SAR)-demining purposes. In addition to the description of the system concept and its requirements, the signal processing steps and the hardware components are presented. Furthermore, the 2D localization accuracy of the system and the classification accuracy of the frequency-coded fingerprints are described in a defined test environment to proof the operational reliability of the realized setup, reaching a classification accuracy of 94.7% and an averaged localization error of 4.9 mm.

## 1. Introduction

Countries affected by war suffer from the aftermath of the weapons used in the contested areas. As a result of the guerrilla warfare in Colombia, for example, numerous active landmines still remain undiscovered and buried underground. In many cases, these mines are improvised explosive devices (IEDs), which mostly contain only small portions of metal. Thus, these mines are challenging to detect with conventional metal detectors. For these purposes, approaches using a ground-penetrating radar (GPR) have been well-established for the contactless detection and imaging of buried objects [1]. Synthetic aperture radar (SAR) algorithms are particularly suitable for taking measures against the weakly reflected signals of the mines, but require the precise determination of the antenna position in order to provide reliable results. The use of global positioning systems (GPSs) provides a low-cost option, but with an accuracy of only 1–3 m, GPS is insufficiently accurate [2,3,4,5]. In order to achieve precise localization, radar system are especially suitable for such tasks. However, the disadvantage of most common higher-dimensional radar-based localization methods is that these systems are very expensive and complex. For this reason, the following paper presents a novel concept that enables higher-dimensional target localizability with the aid of inexpensive and simple hardware extension components. These hardware extensions are based on implementing unique radio frequency (RF) fingerprint codes.

Due to the expansion of wireless communication infrastructure, RF fingerprinting has received significant attention in device identification and localization applications. RF fingerprints result from the physical layer of transmission devices exhibiting imperfections and modulation effects, which can be used to distinguish between specific identifiable features [6]. Thus, the actual differences between hardware components can be reflected in the communication signals of the transmission devices. Furthermore, the extraction of RF fingerprints provides an effective way, for example, to extract additional information about the spatial position of objects in the field of localization methods [7,8,9,10,11,12] and to deal with security threats in wireless communication networks [13,14]. Practical approaches such as those in [15] are based on measuring the received signal strength (RSS). A problem with RSS signatures is that they provide ambiguities because there are several environmental factors that change over time, which may cause a decrease in estimation accuracy [16]. Other research fields are combining pattern recognition algorithms with RF measurements, aiming to enhance the extraction of fingerprint information. However, compared with the nearest neighbor algorithm, no significant improvement can be provided in RSS approaches with the aid of more complex pattern recognition algorithms [17,18]. Regarding the drawbacks of RSS fingerprinting, researchers have proposed different channel-based fingerprinting methods. Time- and frequency-domain considerations such as channel impulse response (CIR) fingerprinting [19,20] or channel transfer function (CTF) fingerprinting [21] have already been investigated. Furthermore, a channel-based autocorrelation in the frequency domain, defined as the frequency coherence function (FCF), was proposed by [22] as a U.S. patent application. In [23], an empirical performance evaluation of RSS-, CIR-, CTF-, and FCF -based location fingerprinting was provided. The results showed that FCF is most robust to channel variations under motion perturbations. Moreover, signal distributions due to unintentional modulation effects of the physical characteristics inside devices and the unpredictable interactions between circuit components were investigated [24]. Third- and forth-order moment (skewness and kurtosis) calculations have been analyzed for feature extraction purposes, leading to good classification perfomance under low SNR and ideal channel conditions. Considering the field of RF fingerprinting for radar applications, recent works have introduced a radar channel frequency coding method in order to actively inject and extract unique fingerprints for the distinction and classification of superimposed subchannel signals [25,26]. With the aid of this method, the cross-range resolution in radar scans can be effectively enhanced due to the extraction of additional information in one-dimensional signals. Based on the frequency coding method, the following paper provides a novel concept for the identification and two-dimensional localization of electromagnetic reflective objects. By using frequency coding filter circuits, we synthesized distinguishable RF fingerprints in radar subchannels in order to spatially classify the measured range profiles and locate handheld systems in order to obtain precise position information for SAR applications. Furthermore, we evaluated the classification and localization accuracy for a moving handheld target.

This paper is organized as follows: in Section 2 the localization principle and its requirements are introduced. Moreover, the signal processing of the corresponding fingerprint extraction and the target classification method are presented. In Section 3, important hardware components such as coding filter, power amplifier, and antennas are proposed and evaluated by their measured performance. The results and analysis are provided in Section 4, and a discussion on the applicability of the system is conducted in Section 5. Finally, the paper is concluded in Section 6.

## 2. System Concept

### 2.1. Localization Principle

The localization principle consists of an FMCW radar, which is used as the signal emitting source. In order to perform a two-dimensional object localization of a handheld (HH) device, an additional wireless station (WS) is used. The WS serves as a secondary communication system, which spans a triangular-shaped area including the FMCW radar and the HH, as can be seen in Figure 1. Furthermore, the calculation of the HH coordinates $x_{h}$ and $y_{h}$ is based on the cosine theorem
(1)$$R_{c}^{2}=R_{a}^{2}+R_{b}^{2}-2R_{a}R_{b}\cos\left(\alpha \right).$$
Here, α is the angle between the FMCW radar and HH system; $R_{a}$, $R_{b}$ and $R_{c}$ are the ranges between the radar, WS, and HH. The range profiles $R_{a}$ and $R_{b}$ and the circumference $R_{d}=R_{a}\;+\;R_{b}+R_{c}$ of the spanned triangle are a direct measure of the radar system.

With these three measures, it is possible to also calculate the range $R_{c}=R_{d}-R_{b}-R_{a}$ between the WS and HH. Because the following relation is valid
(2)$$\cos\left(\alpha \right)=\frac{x_{h}}{R_{a}},$$
equalizing (Equation 1) and (Equation 2) leads to
(3)$$x_{h}=\frac{R_{a}^{2}+R_{b}^{2}-R_{c}^{2}}{2R_{b}}.$$

With the help of the Pythagorean theorem, $y_{h}$ can be calculated as follows
(4)$$y_{h}=\sqrt{R_{a}^{2}-x_{h}^{2}}.$$

In this context, the exact 2D position of the HH device can be determined in only one FMWC radar measurement. The complete system set-up and the distinction of the different range profiles are explained in the following sections.

### 2.2. System Set-Up and Requirements

As explained in Section 2.1, the localization principle is based on an FMCW radar system, a WS, and a HH device. Regarding the frequency modulation
(5)$$\omega \left(t\right)=\omega_{0}+\dot{\omega }t$$
of the transmitted signal, its time-dependent phase ϕ(t) can be expressed as follows: (6)$$\phi \left(t\right)=\int_{0}^{t}\omega \left(t^{\prime }\right){dt}^{\prime }=\omega_{0}t+\dot{\omega }\frac{t^{2}}{2}+\phi_{0}$$
where the $\dot{}$ operator indicates the temporal derivative of the frequency ω. Thus, the signal model of the transmitted signal $s_{T}\left(t\right)$ is given as
(7)$$s_{T}\left(t\right)=\hat{s}\cos\left(\omega_{0}t+\dot{\omega }\frac{t^{2}}{2}+\phi_{0}\right)$$
where $\hat{s}$ is the amplitude of the signal. Due to channel modulation effects, the time-delayed receiving signal can be expressed as a convolution of the transmitted signal and the corresponding CIR h(t)
(8)$$s_{R}\left(t\right)=s_{T}\left(t-\tau \right)*h\left(t\right).$$
where ∗ is the convolution operator. The parameter τ is the time of flight and can be extracted in the signal-processing step after mixing the transmit and receive signal. The parameter is defined by
(9)$$\tau =\frac{2R}{c}$$
where *c* is the light speed constant, and *R* is the line-of-sight (LOS) distance to the target.

So far, an ideal environment is assumed, where each subchannel inhibits the same characteristics. Because there are three range profiles ($R_{a}$, $R_{b}$ and $R_{c}$) in this work, which are not distinguishable without further system adaption, each subchannel of $s_{T}$ needs to exhibit a specific fingerprint in order to map the range profiles to their location in the set-up. Here, these fingerprints are inserted by adding different bandstop filters in the transmission lines of each corner point of the triangular-shaped measurement area. Adding different bandstop filters to the signal path then leads to distinguishable subchannels in which each channel possesses different CIR’s $h_{i}$ or CTF’s $H_{i}$. As can be seen in Figure 2, each CTF features frequency-coded information, indicated by different bandstop notches. Here, the range $R_{a}$ between the radar and HH device corresponds to the coded information of the CTF of $H_{1}$, and the range $R_{b}$ between FMCW and WS corresponds to $H_{2}$. Because the range $R_{c}$ is an indirect measure calculated by subtracting $R_{a}$ and $R_{b}$ from the measured circumference of the spanned triangle, its CTF corresponds to both $H_{1}$ and $H_{3}$. By passing through both filters, however, no ambiguities appear, because the coding is singularly assigned to the measured circumference.

In this context, the block diagram in Figure 3 shows the set-up adaptation of the localization principle, with all the required hardware components

The FMCW radar is used to transmit and receive its signal to two different directions. Here, antenna ${\mathrm{Ant}}_{1}$ receives the reflection of the HH system, which is assigned to the range profile $R_{a}$. In order to scan a huge area in which the HH device can be located, ${\mathrm{Ant}}_{1}$ needs a wide beam angle. In order to achieve a suitable compromise between the antenna gain and scan area, both antennas $Ant_{1}$ and $Ant_{2}$ should provide beam angles between 30° and 70°. The frequency coding of the range profile $R_{a}$ is performed with the aid of the bandstop filter $H_{1}$. To prevent unnecessary losses over the bandwidth of the transmitted signal, the notch frequency of $H_{1}$ needs to be very narrow.

Antenna ${\mathrm{Ant}}_{2}$ is then used to send the signal to antenna ${\mathrm{Ant}}_{3}$, which is connected to the WS. Because the FMCW radar and WS are stationary, the beam angle of both antennas can be chosen to be very narrow in order to achieve a high gain factor. After receiving the signal from the FMCW radar, the WS couples the signal into two directions. The bandstop filter $H_{2}$ is used to reflect the signal back to the FMCW radar and to perform the frequency coding of the range profile $R_{b}$. Here, the reflection characteristic of the filter can be realized by a short-circuited port. The second coupling path of the WS is then used in order to redirect the signal to the HH system via antenna ${\mathrm{Ant}}_{4}$, which holds the same antenna specifications as ${\mathrm{Ant}}_{1}$ for reasons of symmetry. Due to the signal coupling, it is important that the power divider is perfectly matched to prevent further power losses. In order to decrease the lack of power due to signal coupling, an additional power amplifier (PA) is used. The bandstop filter $H_{3}$ then performs the frequency coding for the measured circumference ($R_{d}=R_{a}+R_{b}+R_{c}$). Furthermore, the HH device is considered to be a strong reflecting object.

### 2.3. Fingerprint Extraction and Target Classification

As explained in the previous sections, distinguishing the range profiles requires coding filters in order to inject unique fingerprints into each radar echo. Figure 4 shows the corresponding signal processing steps for the extraction and classification of the frequency-coded radar echo information.

Here, $s_{\mathrm{IF}}\left(t\right)$ is the low-pass-filtered signal of the mixture of the transmit signal $s_{T}$ and the receive signal $s_{R}\left(t\right)$. First, a short-term Fourier transformation (STFT) is calculated for the measured signal $s_{\mathrm{IF}}\left(t\right)$.
(10)$$S_{\mathrm{IF}}\left(\omega ,\tau \right)=\int_{-\infty }^{\infty }s_{\mathrm{IF}}\left(t\right)w\left(t-\tau \right)e^{-j\omega \tau }dt$$

The STFT determines which fingerprint occurs at which target and thus represents the time and frequency components of a radar echo. Due to the two-dimensional representation of a one-dimensional signal, time and frequency components are limited in their resolution. The general statement is that the time duration Δt and bandwidth Δω for any signal are defined by the following relation
(11)Δt·Δω≥2π.

Because the signal is segmented in the STFT by applying a window w(t−τ) to it, the limiting element is the size of the corresponding window function [27]. The window size thus determines the resolution to distinguish two targets and the precision for the extraction of the implemented fingerprint features.

In order to classify the radar echos, each echo needs to be initially time-gated in the next processing step. Due to the frequency notch characteristics of the injected fingerprint, the direction of the frequency gradients at each time scale (here, range scale) can be considered as a classification criterion. Furthermore, the direction-based gradient consideration is a valid choice because real-world measurements of signal fingerprints are affected by their environment and the composition of the targets. Figure 5 shows a time-gated echo fingerprint of a radar echo $R_{a}$ within the X-band, which exhibits a bandstop notch at approximately 9.81 GHz. As can be seen, the fingerprint features become fuzzy along the time scale. Due to the nature of the bandstop filter, the negative gradients along the range axis are more likely to persistently occur in the direction of the injected bandstop notch frequency (see Figure 5b). Gradients in addition to the frequency band of the bandstop resonances are more likely to change their direction over the range scale (see Figure 5c). In order to extract persistent gradients along the time scale, a filter matrix $\omega_{g}$ is correlated with the frequency gradients of the signal $S_{\mathrm{IF},\mathrm{Gated}}$. The size (number of matrix rows) of the corresponding gradient filter $\omega_{g}$ depends on the number of frequency points of $S_{\mathrm{IF},\mathrm{Gated}}$ and has the following form: (12)$$\omega_{g}=\left[\begin{array}{ccc} 1 & 0 & -1\\ \vdots & \vdots & \vdots \\ 1 & 0 & -1 \end{array}\right].$$

Finally, the filtered features are summed up and normalized, such that a probability of feature occurrences can be analyzed and used for the feature classification.

In short, the measured signal is first transformed into the time–frequency domain and the detected targets are time-gated. After that, the corresponding fingerprints of the time-gated targets are filtered out based on their frequency gradients and are finally classified.

## 3. Hardware Design and Measurement

For the construction of the system, various hardware components are needed. First, the bandstop filter circuit and its corresponding measurements are described. In order to ensure a strong wireless connection between ${\mathrm{Ant}}_{2}$ and ${\mathrm{Ant}}_{3}$, horn antennas were chosen. Vivaldi antennas were chosen for the design of ${\mathrm{Ant}}_{1}$ and ${\mathrm{Ant}}_{4}$ due to their wide-beam characteristics. Both antenna designs were evaluated by measurements of the corresponding input reflection and beam characteristics. Additionally, a PA circuit is presented, which is needed for the redirection purpose in the WS.

### 3.1. Bandstop Filter

The manufactured circuit of one of the three bandstop filters ($H_{1}$, $H_{2}$ and $H_{3}$) can be seen in Figure 6. The substrate material is RO4003 from Rogers Corporation (Chandler, USA). The design is based on two $\frac{\lambda }{2}$-lines, which are destructively coupled with a $\frac{\lambda }{4}$-phase shift line [25,28,29]. The destructive interferences of the phase shift line and the coupling lines result in a notch characteristic on a narrow frequency-scale.

As can be seen in Figure 7a, notch characteristics below −6 dB were achieved for both the transmission ($H_{1}$, $H_{3}$) and reflection filter ($H_{2}$). Because the design of the complete system must have the highest possible matching for the many coupled paths, Figure 7b shows the input and output reflection measurements of the transmission filter. As can be seen, both port reflections are below −10 dB over the entire bandwidth, which ensures a good matching. These characteristics show that the filter design is a good match with the system requirements.

### 3.2. Antennas

The chosen structure and dimensions for the design of the horn antenna are based on [30].

Due to cost and mobility reasons, the material of the manufactured antenna shown in Figure 8 was chosen to be aluminum. The Vivaldi antenna was manufactured with RO4003 substrate from Rogers Corporation (Chandler, USA), and is shown in Figure 9 [31,32]. For the measurement of the horn antenna, an SMA waveguide adapter was attached to the WR90 standard flange of the antenna. Because both antennas require high input matching, Figure 10 shows the respective reflection measurement $S_{11}$ of the antennas.

As can be seen, a good matching was achieved for both antennas, which was below −11 dB over the entire bandwidth. In addition to the S parameters, the directional diagrams of both antennas were recorded for the electric- and magnetic-plane distributions. The measurements for a center frequency of 10 GHz are shown in Figure 11 for the E field and in Figure 12 for the H field. It can be seen that the directional diagram of the horn antenna’s E field has a full width at half maximum (FWHM) of 16°. The directional diagram of the H field has an FWHM of 15°. These results show that a good directional connection between the radar and WS could be achieved due to the focused beam characteristics. The directional diagrams of the Vivaldi antennas E field show an FWHM of 48°. The directional diagram of the corresponding H field shows an FWHM of 70°. These results show that the radiation of the Vivaldi antenna has a sufficiently large angle. Thus, large areas can be measured in which the HH device can be detected.

### 3.3. Power Amplifier

For the PA circuit, a low-noise amplifier (LNA) QPA2609 from Qorvo (Greensboro, NC, USA) was used. The manufactured circuit can be seen in Figure 13. The respective S-parameter measurements are shown in Figure 14. As can be seen, the input and output reflections were sufficiently low. Furthermore, a forward transmission amplification gain of approximately 20 dB and a reverse transmission attenuation of approximately −60 dB were achieved, indicating the good redirection characteristics of the WS.

## 4. System Validation

In order to validate the system concept, the fingerprint classification and localization accuracy are separately evaluated in the following sections. For the measurements, a vector network analyzer (VNA) ZVA40 from Rohde & Schwarz (Munich, Germany) was used in order to record the radar data. Furthermore, the HH device was installed on a motor-driven positioning system using a step-controller iMC-S8 from isel in order to compare the RF measurements with ground truth data. To allow classification accuracy and localizability to be separately considered, both available channels of the VNA were used instead of a power divider in the transmit and receive paths of the radar device. The corresponding measurement set-up is shown in Figure 15.

### 4.1. Classification Accuracy

For the determination of the classification accuracy, the classified signals were compared to the a priori information of their true classes. Figure 16 shows a recorded range profile of the superimposed signals.

As can be seen, the distance of the WS ($R_{b}$) was farther than the triangle circumference ($R_{d}$). This resulted from the fact that the links of the respective stations contained a cable offset, because the respective pairs of the antennas of the radar and WS could not be assigned to the same corner point of the triangle. Furthermore, this offset was used to artificially lengthen the directional link between the radar and WS. The artificial extension of the cable length offers two advantages here: First, the effect of the reflection bandpass $H_{2}$ can be decoupled from the effects of the antenna ${\mathrm{Ant}}_{3}$ to obtain more precise fingerprint features. Second, it allows the HH and WS detections to be clearly separated in the range profile to avoid ambiguities of the targets. Thus, different areas of the assumed HH position can be adjusted for different applications to achieve a more precise localization. According to the range plot, the three targets were analyzed, as described in Section 2.3. Because the classification is strongly influenced by the parameters of the STFT, the parameter choices are shown in Table 1.

According to this, Figure 17 shows the time-gated plots of the range–frequency representation for the ranges $R_{a}$, $R_{b}$, and $R_{d}$. After the signal-processing steps, the extracted and normalized probabilities of the feature occurrences were obtained, as shown in Figure 18.

As can be seen, the features for ranges $R_{a}$ and $R_{b}$ could be extracted very well without other frequency-band interferences leading to ambiguities of the classification. For $R_{c}$, there were interferences, which were small in comparison with the features of interest. For a sum of 44 localization measurements, Table 2 shows the classification accuracy of the single ranges.

As can be seen, the gradient-based filtering method leads to good classification results for the injected RF fingerprints.

### 4.2. Measurement Accuracy

To verify the measurement accuracy of the system, the distances $R_{a}$, $R_{b}$, and $R_{c}$ were used to calculate the position of the HH. While the HH was mounted on a motor driven positioning system and moved along a given path, its ground truth was used to calculate the the measurement error of the system. Because the antennas of the radar system are located on different coordinate points, a systematic measurement error occurs. The same holds for the antennas of the WS. As can be seen in Figure 19, the offset lengths Δa, $\Delta b_{1,2}$, and Δc need to be considered in order to correctly localize the HH.

In this context, the HH coordinate calculations change, as follows
(13)$$x_{h,\mathrm{meas}}=\frac{{\left(R_{a,\mathrm{meas}}+\Delta a\right)}^{2}+{\left(R_{b,\mathrm{meas}}+\Delta b_{1}+\Delta b_{2}\right)}^{2}-{\left(R_{c,\mathrm{meas}}+\Delta c\right)}^{2}}{2(R_{b,\mathrm{meas}}+\Delta b_{1}+\Delta b_{2})}.$$
(14)$$y_{h,\mathrm{meas}}=\sqrt{{\left(R_{a,\mathrm{meas}}+\Delta a\right)}^{2}-x_{h,\mathrm{meas}}^{2}}.$$

The calibration of the offset lengths were iteratively calculated. Here, the error between the ground truth and measurement was minimized by iterative changes in the cable lengths. Due to the determination of the offset parameters, the measured path of the HH could be reconstructed. Figure 20 and Figure 21 show the uncalibrated and calibrated locations of the HH device, respectively.

As can be seen, the offset of the cable lengths leads to a tilting of the square area of the true HH location. After the calibration of the cable lengths, a good match of the measured HH location compared with its ground truth was achieved. For the calculation of the average location error, the Euclidean distance was chosen to be the distance measure between the reference and measured locations. The mean error was 4.9 mm, which shows that a high precision of the HH system location can be achieved.

## 5. Discussion

As can be seen from the results in Section 4, a very precise position determination for HH devices could be achieved with an average error of 4.9 mm. Comparing this with other methods in the field of fingerprinting localization, the authors in [8] proposed an indoor localization method using fingerprint feature extraction with bluetooth low-energy beacon fingerprints. However, this method has an average error of 680 mm and can only be used for indoor applications. In [9], a method was introduced that can also be applied in dynamic environments in outdoor applications. Here, the authors proposed a novel positioning framework based on multiple transfer learning fusion using generalized policy iteration. However, the average error that could be obtained with this method was 540 mm in the best test scenario. Regarding this, our presented concept has a much higher localization accuracy than other methods in the field of fingerprinting localization and is applicable for both indoor and outdoor environments undergoing dynamic environment changes. However, a trade-off associated with the measurement principle arises due to the uncertainty relationship in the time–frequency domain. Because high-resolution temporal behavior is needed to obtain a high classification accuracy of the RF fingerprints, the range resolution of the radar suffers. Thus, it must be assumed that the three targets to be detected (HH, triangular circumference, and WS) need to have sufficient range distances from each other in order to avoid classification errors. An overlap of the distance profiles means an overlap of the fingerprints, which can therefore lead to ambiguities. Additionally, misclassifications have a direct influence on the accuracy of the position determination. Furthermore, the structure of the measurement principle results in cable lengths that must first be calibrated for correct applicability. With a static set-up, however, this calibration must only be performed once and not for each new measurement. Furthermore, calibration represents a critical point for the measurement accuracy of the system. This is mainly due to the fact that a calibration error not only leads to rotation of the underlying coordinate grid, but also to tilting. An improvement in the measurement accuracy may be achieved by an increase in the calibration points to reduce the mean square error. In addition to the calibration, the reflection properties of the HH contribute to a significant error. As soon as the HH has a nonuniform reflection to the WS and radar, this leads to a range offset. To reduce this error term to a minimum, antennas with a high beam angle of about 180° should be used to achieve an almost orientation independent reflection of the HH. Furthermore, the previous considerations of 2D localization can be extended to a 3D localization method. In order to consider another dimension for 3D localizability, another WS would be needed that additionally contains a distinguishable fingerprint filter. This second WS has to be located at the same position as the original WS and has to exhibit a known vertical distance to it. Thus, the height information to the WS and thus to the HH can be extracted.

## 6. Conclusions and Future Work

In this study, we examined the usability of RF fingerprints for the applicability of a novel radar-based localization concept. The concept makes use of frequency-coding filters that modify RF signals in such a way that differentiable subchannel features can be distinguished from each other. The distinction is based on the extraction of gradient features in the time–frequency domain using the STFT method. Compared with conventional RF fingerprinting approaches, no prior information or large data sets about the measurement environment are required. To investigate the concept, various hardware components, such as bandstop filters, horn antennas, Vivaldi antennas, and a PA, were introduced and evaluated based on their measurements. In addition, in a defined measurement environment, by calculating the error between the localization measurement to its ground truth, we showed that a high localization accuracy with an average error of 4.9 mm could be achieved. It was also shown that the gradient-based feature extraction method achieved an average classification accuracy of 94.7%. Nevertheless, due to the time–frequency analysis, there is a trade-off between target resolution and feature identification. Regarding this, our future work will focus on enhancing the impact of the transformation methods on the target resolution. Furthermore, possible improvements to the fingerprint insertion by increasing quality factors of the filters will be analyzed. 

## Figures and Tables

**Figure 1 sensors-22-08688-f001:**
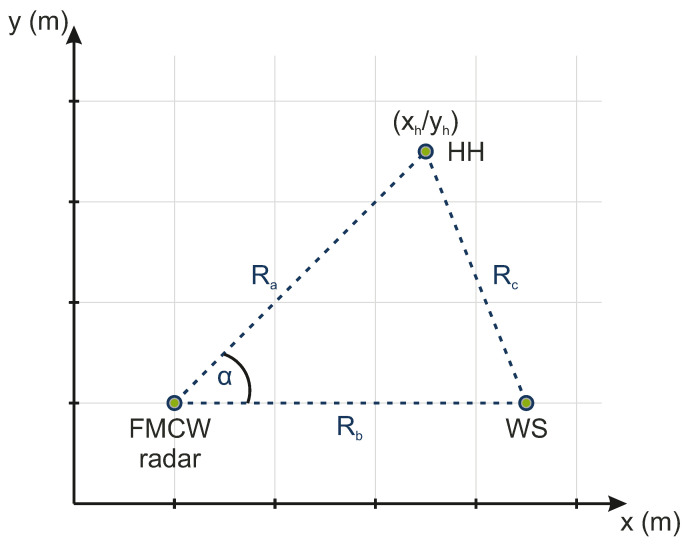
Localization principle including a radar system, WS, and HH device, which are located within distances $R_{a}$, $R_{b}$, and $R_{c}$ from each other. The position of the HH device to be located is defined at the coordinate point ($x_{h}/y_{h}$).

**Figure 2 sensors-22-08688-f002:**
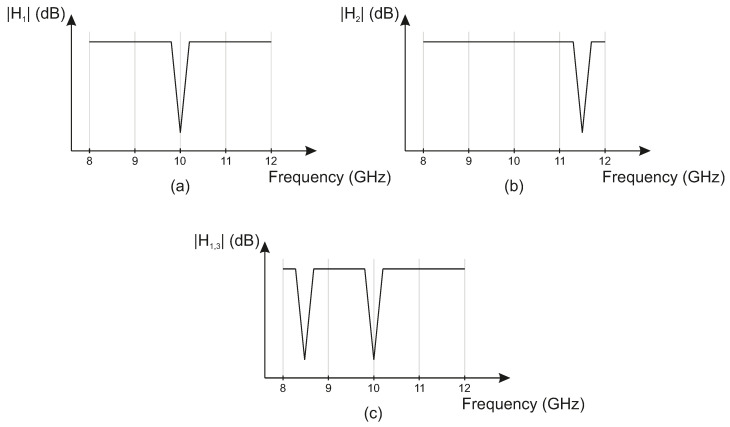
Transfer functions $|H_{1}|$, $|H_{2}|$, and $|H_{3}|$ (**a**–**c**) for distinguishing ranges $R_{a}$, $R_{b}$ and $R_{c}$ due to distinguishable frequency codes (bandstop notches).

**Figure 3 sensors-22-08688-f003:**
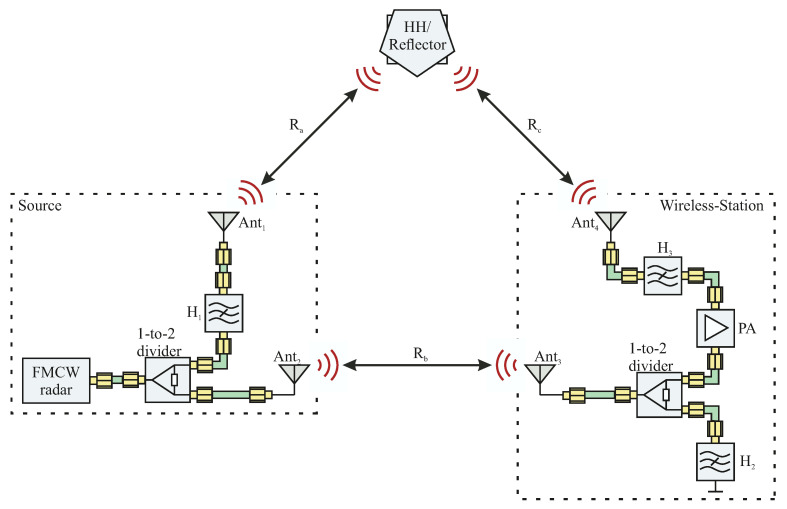
Set-up adaptation of the localization principal including the coding filter $H_{1}$, $H_{2}$, and $H_{3}$; the transmit/receive antennas ${\mathrm{Ant}}_{1}$, ${\mathrm{Ant}}_{2}$, ${\mathrm{Ant}}_{3}$, and ${\mathrm{Ant}}_{4}$; a 1-to-2 power divider; and a PA.

**Figure 4 sensors-22-08688-f004:**
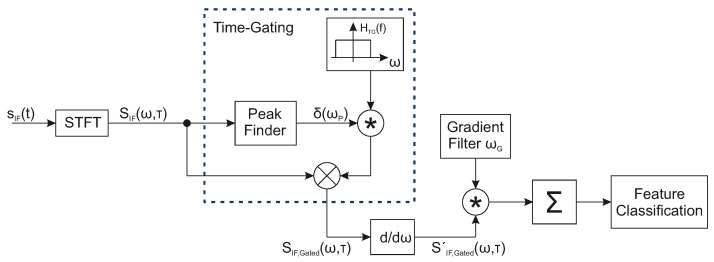
Signal flow diagram of the fingerprint extraction and radar echo classification.

**Figure 5 sensors-22-08688-f005:**
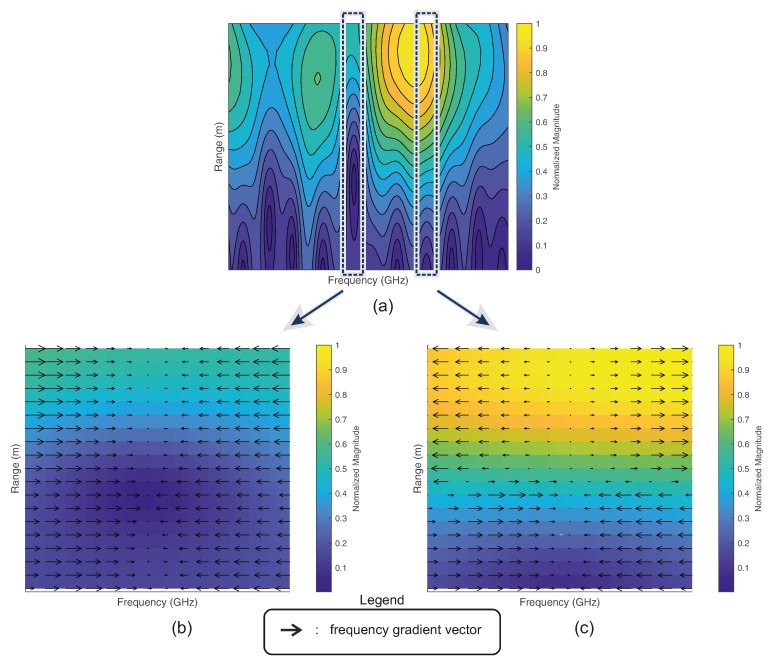
Time-gated fingerprint of a detected target (**a**) and corresponding frequency-gradient vectors at the notch frequency (**b**), which persistently occur along the range axis in the direction of the notch and the corresponding frequency gradient vectors outside the notch frequency (**c**), which change their direction along the range axis. The orientation of the arrows corresponds to the direction of the frequency gradient, and the length of the arrows corresponds to the magnitude of the frequency gradient.

**Figure 6 sensors-22-08688-f006:**
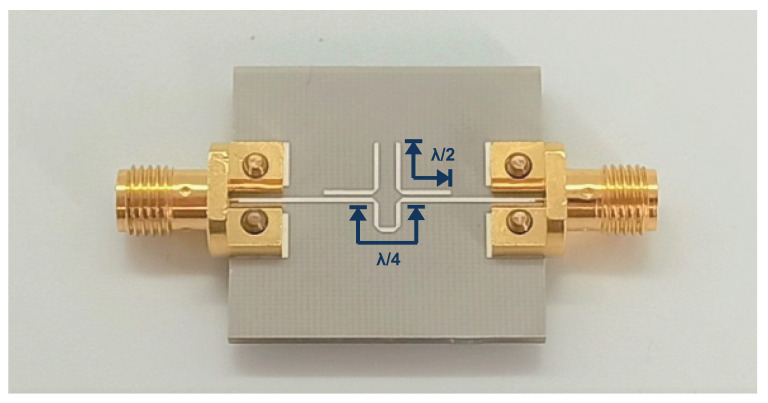
Manufactured bandstop filter circuit with end-launch SMA connectors.

**Figure 7 sensors-22-08688-f007:**
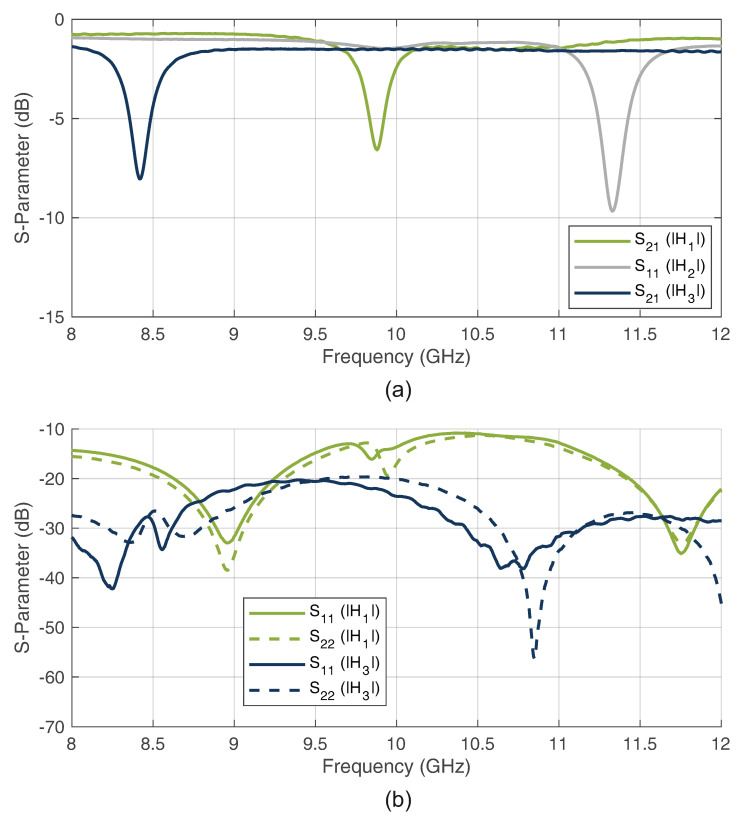
Measured S-parameters of the proposed bandstop filter notch properties (**a**) and the filter matching (**b**).

**Figure 8 sensors-22-08688-f008:**
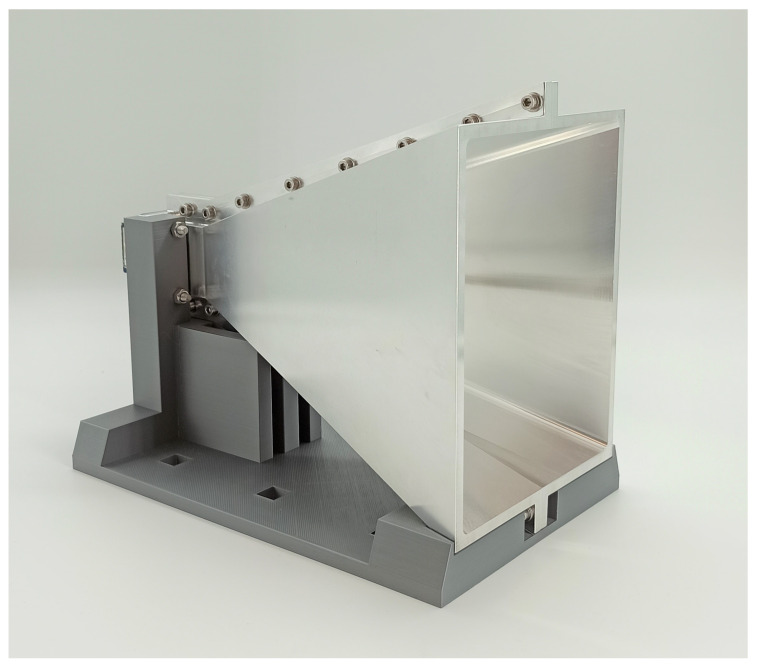
Manufactured horn antenna.

**Figure 9 sensors-22-08688-f009:**
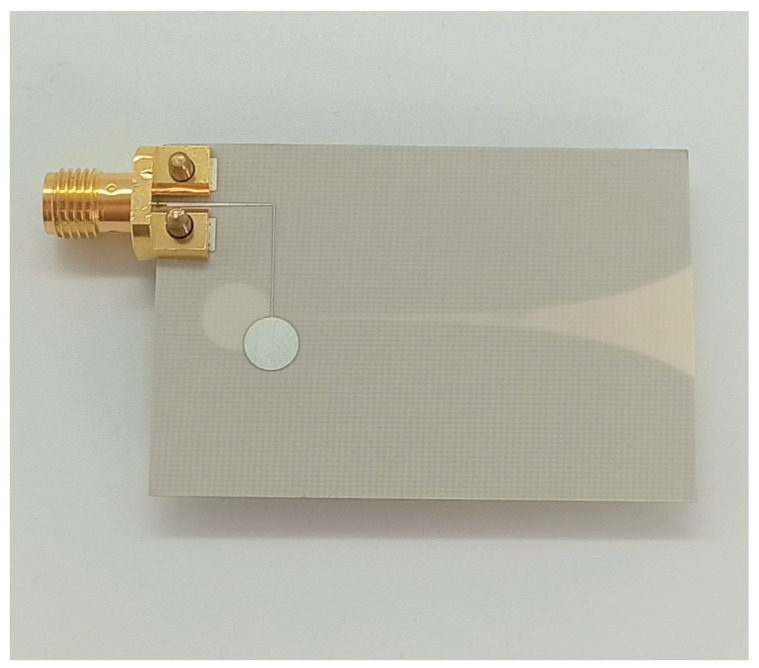
Manufactured Vivaldi antenna.

**Figure 10 sensors-22-08688-f010:**
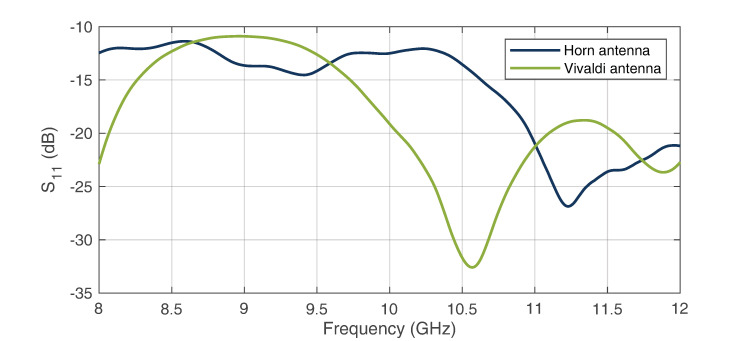
$S_{11}$ measurements of the proposed horn and Vivaldi antennas.

**Figure 11 sensors-22-08688-f011:**
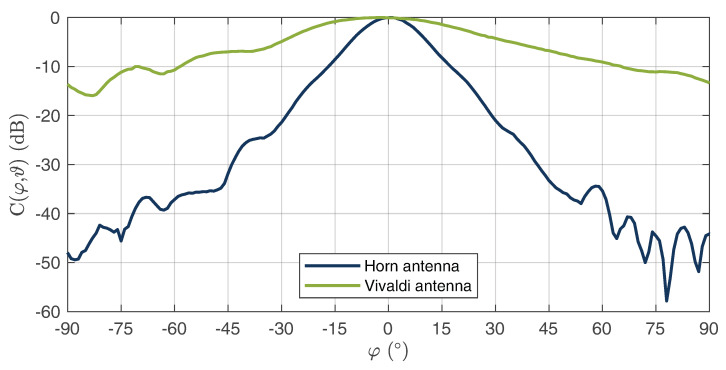
Measured E-field radiation pattern of the horn and Vivaldi antennas at 10 GHz.

**Figure 12 sensors-22-08688-f012:**
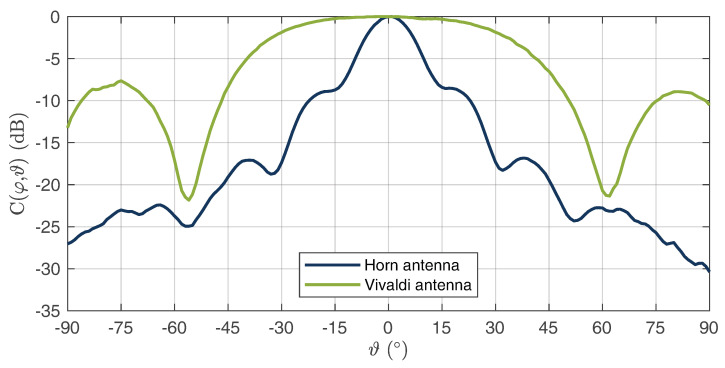
Measured H-field radiation pattern of the horn and Vivaldi antennas at 10 GHz.

**Figure 13 sensors-22-08688-f013:**
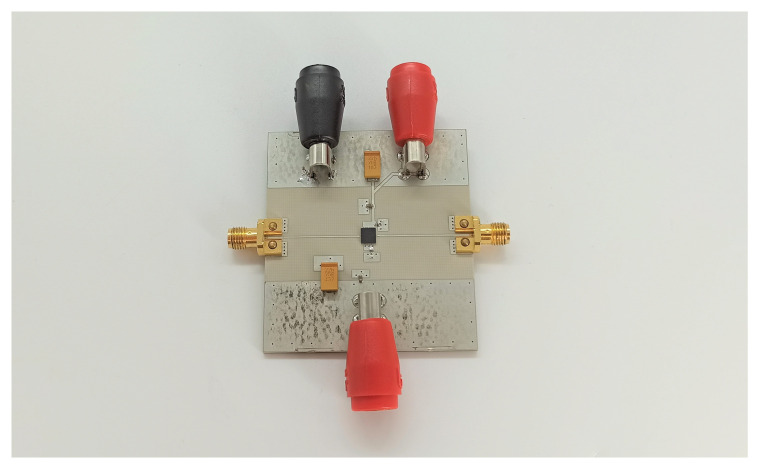
Manufactured PA circuit.

**Figure 14 sensors-22-08688-f014:**
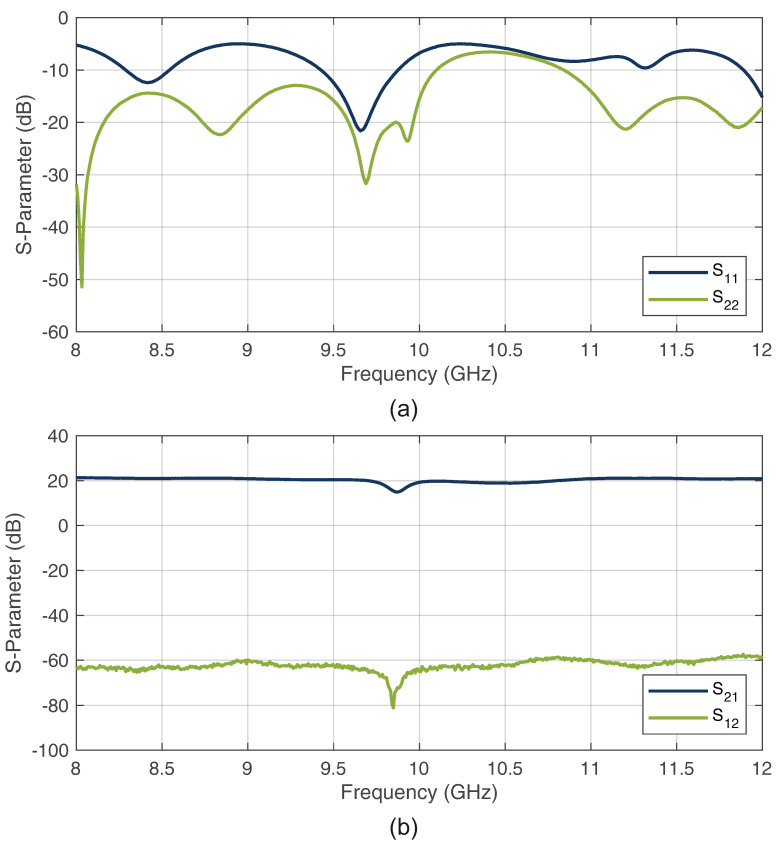
S-parameter of the reflection measurements (**a**) and the transmission measurements (**b**) for the proposed PA.

**Figure 15 sensors-22-08688-f015:**
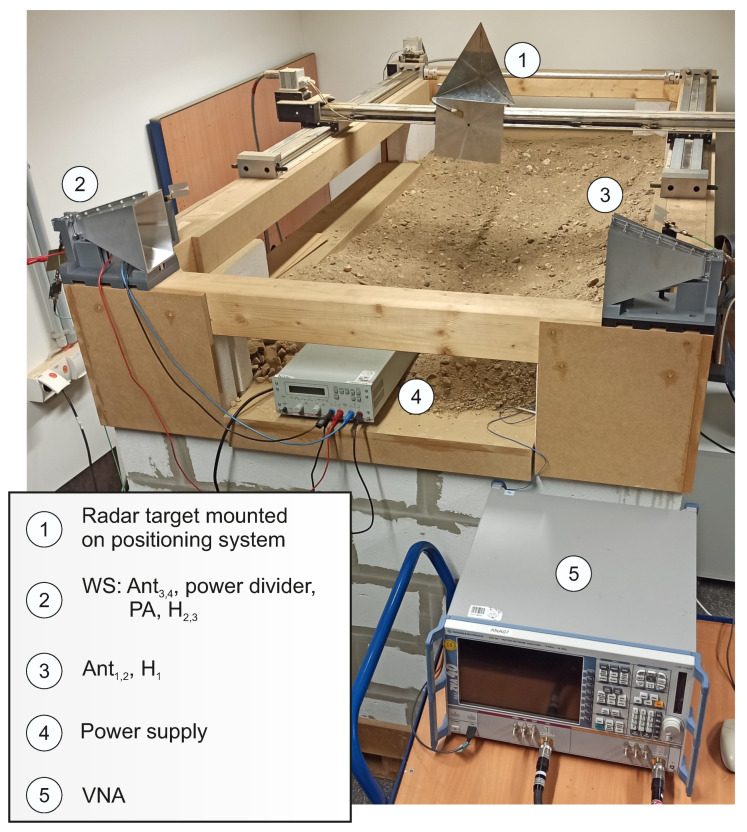
Measurement set-up of the localization principle consisting of a VNA as radar device, WS, and electromagnetic reflective objective as the HH device mounted on a motor-driven positioning system.

**Figure 16 sensors-22-08688-f016:**
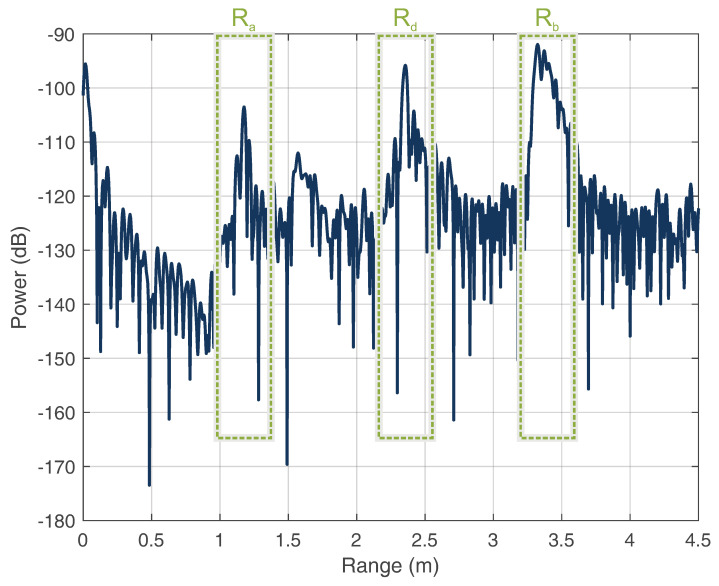
Range plot of the set-up showing the radar echos of the HH ($R_{a}$), WS ($R_{b}$), and echo of a triangle circumference ($R_{d}=R_{a}+R_{b}+R_{c}$).

**Figure 17 sensors-22-08688-f017:**
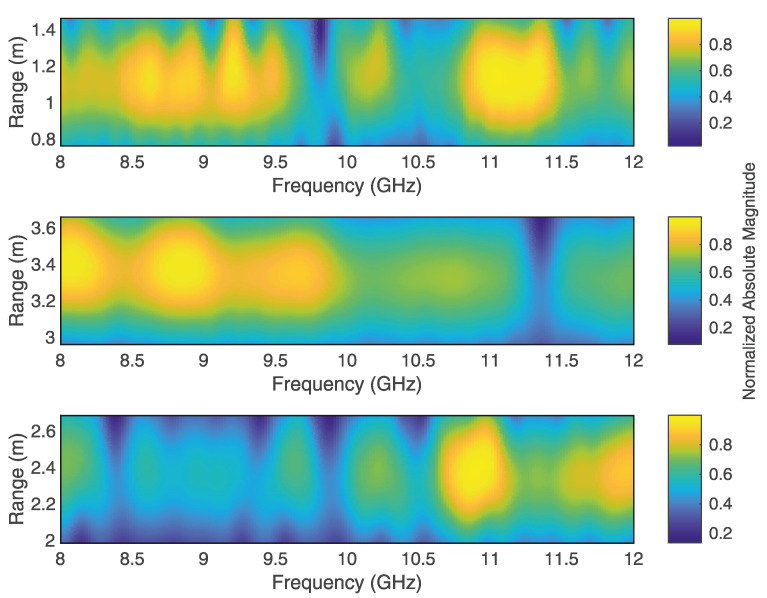
Time-gated STFT representations of the measured radar echos ($R_{a}$, $R_{b}$ and $R_{d}$).

**Figure 18 sensors-22-08688-f018:**
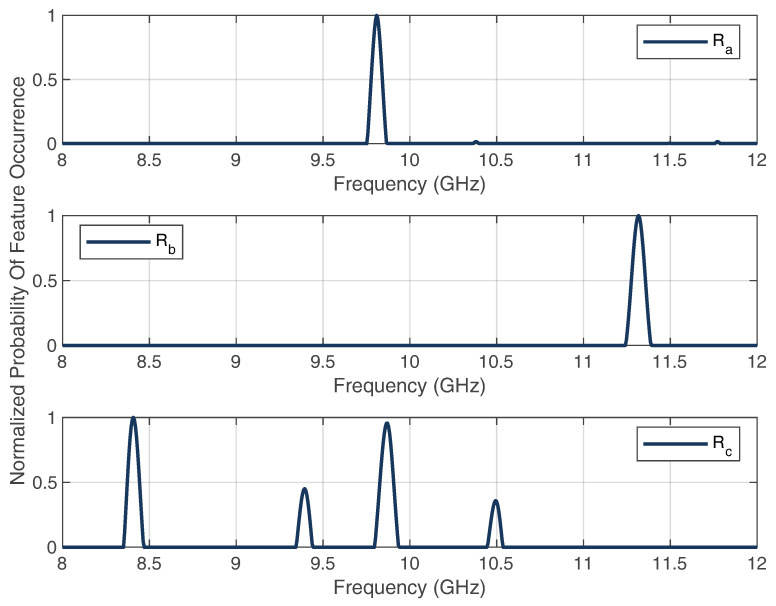
Extracted and normalized probability of the features occurrences for the range distances $R_{a}$, $R_{b}$, and $R_{d}$.

**Figure 19 sensors-22-08688-f019:**
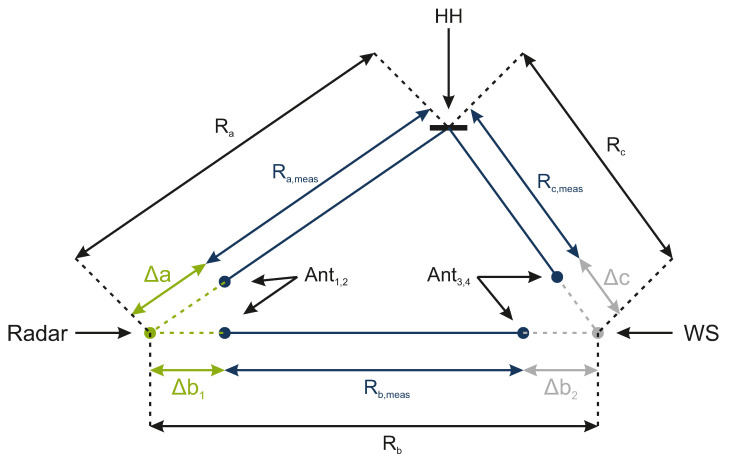
Schematic diagram of the localization error for the HH’s position due to dismissed cable offset length between the antennas.

**Figure 20 sensors-22-08688-f020:**
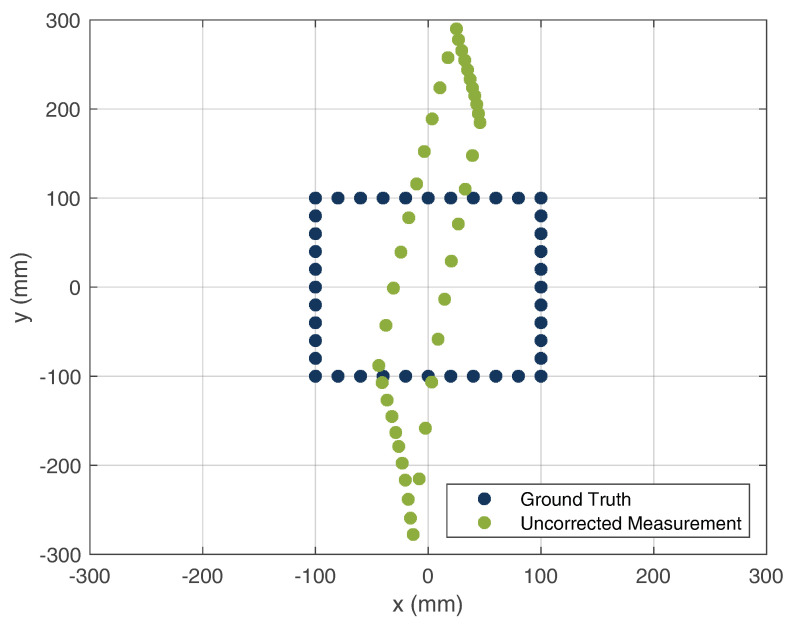
Uncorrected localization measurement compared with the ground truth of the HH position.

**Figure 21 sensors-22-08688-f021:**
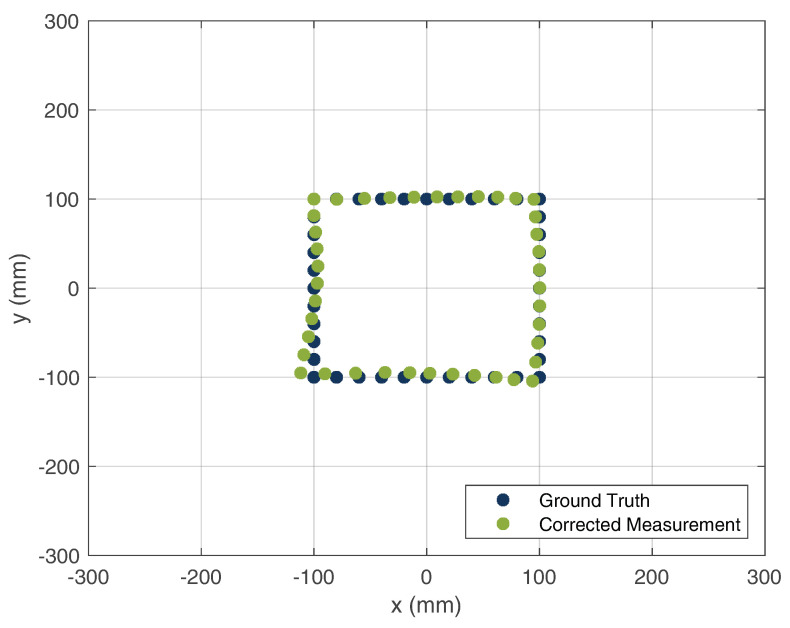
Corrected localization measurement compared with the ground truth of the HH position.

**Table 1 sensors-22-08688-t001:** Chosen parameter for the STFT of the radar measurements.

Parameter	Value
Window size	65
Window overlap	64
Number of sampling points	$2^{14}$
Window function	Hanning-Window

**Table 2 sensors-22-08688-t002:** Classification accuracy of the correctly assigned RF fingerprints to the measured distances $R_{a}$, $R_{b}$, and $R_{d}$.

Range	Value
$R_{a}$	97.73%
$R_{b}$	100%
$R_{d}$	86.36%

## Data Availability

Not applicable.

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
