# Peer review of "A Novel Localization System in SAR-Demining Applications Using Invariant Radar Channel Fingerprints"

_sensors, 2022, doi:10.3390/s22228688_

Round 1
Reviewer 1 Report
This manuscript present a novel two dimensional frequency modulated continous wave (FMCW) radar localization method for handheld systems based on the extraction of distinguishable subchannel fingerprints. Especially, I agree to have new method to design a localization system in demining. However, here are some comments to improve paper further better.
1. Abstract
The author described the main work of this study, but the content is brief. It is recommended that the author briefly describe the background or significance of the study, and what problems are designed to solve.
References:
A.Marsh, L., Van Verre, W., L. Davidson, J., Gao, X., JW Podd, F., J. Daniels, D., & J. Peyton, A. (2019). Combining electromagnetic spectroscopy and ground-penetrating radar for the detection of anti-personnel landmines. Sensors, 19(15), 3390.
Fernández, R., Montes, H., & Armada, M. (2016). Intelligent multisensor prodder for training operators in humanitarian demining. Sensors, 16(7), 965.
2. Introduction
The manuscript described the research progress of radio frequency fingerprinting (RF), received signal strength (RSS) and frequency coding,. However, the first paragraph is slightly lengthy, so it is recommended that the author divide it into two paragraphs, and focus on sorting out the research progress of SAR on demining applications. In addition, it is necessary to summarize what problem this study is addressing.
3. System Concept
“2.2 System Setup and Requirements”, in order to s scan a huge area and get a high gain factor, the author should clarify the specific values of the beam angle and notch frequency corresponding to Ant1 and H1.
4. Hardware Design and Measurement
The author describe the Bandstop Filter, Antennas and Power Amplifier. Because the study by Francesco Soldovieri et al. pointed out that FMCW Radar need to consider the antenna effect and the interaction of the soil dielectric properties and the target, did the authors consider the influence of these factors when designing and measuring the hardware?
References:
F.Soldovieri, O. Lopera and S. Lambot, "Combination of Advanced Inversion Techniques for an Accurate Target Localization via GPR for Demining Applications," in IEEE Transactions on Geoscience and Remote Sensing, vol. 49, no. 1, pp. 451-461, Jan. 2011, doi: 10.1109/TGRS.2010.2051675.
5. System Validation
The author evaluate the classification accuracy and measurement accuracy of the system. Has the author applied the method to the location of mines such as AT, PMN2 and M14?
6. Discussion
In addition to summarizing the results of Chapter 3, the author should discuss the validation results of Chapter 4, and analyze the reasons that affect the accuracy of the evaluation.
Author Response
Dear Reviewer,
Thank you very much for your review and remarks. Please find attached a detailed answer to your questions

Reviewer 2 Report
The authors presented a novel two-dimensional FMCW localization method for handheld systems based on the extraction of distinguishable subchannel fingerprints. The paper looks well organized and well-written, but I have some comments and suggestions as follows:
-The authors should write the system accuracy percentage in the last sentence in the abstract.
- The paper's contributions are not clear in the introduction. I suggest to breakdown the contribution into several points before the paper organization.
- In the last paragraph of the introduction The paper's organisation is not presented well. Section 2 presents the system concept not, important hardware components. Section 3 presents Hardware Design and Measurement, not the results and analysis. Section 4 discusses the system validation. Please check them all and rewrite them properly.
- I suggest adding a new section to discuss the most recent related works.
- In section 4, System Validation, I did not see any simulation results or even a comparison of the proposed design with other designs from existing works. Besides, what is the system accuracy and how is your proposed design achieve better accuracy compared to others?
- Section 5 is very short. I suggest merging Section 5 with the previous section.
Author Response

(The authors gave the same response as above.)

Round 2
Reviewer 1 Report
Thank you very much for the time you spend for revising the manuscript, and there are some suggestions for figures in this manuscript.
Line142-145, Figure 4 shows the corresponding signal processing steps for the extraction and classification of the frequency coded radar echo information. It is recommended that author should briefly summarize the steps and contents contained in this figure.
Figure 5 showed the frequency gradient vectors change their direction along the range axis at notch frequencies (b) and (c). Among them, the value of arrows with different lengths should be different, so it is recommended that author add a legend to label the values of the arrows.
Reviewer 2 Report
Thanks for addressing my concerns, but I still have some points that need to be addressed before the final publication as follows:
-Again, the paper's organization is not described well, please check the section numbering. The introduction should be numbered 1.
- In the introduction, you discussed several indoor localizationsystem based on fingerprint techniques (i.e. [6-8]), you also should include some works such as: "Robust 3D indoor positioning system based on radio map using Bayesian network" and "Improving accuracy in indoor localization system using fingerprinting technique "A three-dimensional pattern recognition localization system based on a Bayesian graphical model".
- In this Discussion "As can be seen from the results in chapter 3, a very precise position determination 282 for HH devices can be achieved" What is chapter 3? This is an article, not a thesis.
